# A Therapeutic Pathway in Patients with Chronic Coronary Syndromes: Proposal for Optimization

**DOI:** 10.3390/jcm11082091

**Published:** 2022-04-08

**Authors:** Raffaele De Caterina, Paolo Calabrò, Gianluca Campo, Roberta Rossini, Simona Giubilato

**Affiliations:** 1Chair of Cardiology, University of Pisa, 56126 Pisa, Italy; 2Cardio-Thoracic and Vascular Department, Azienda Ospedaliero-Universitaria Pisana, 56124 Pisa, Italy; 3Fondazione VillaSerena per la Ricerca, 65013 Città Sant’Angelo, Italy; 4Department of Translational Medical Sciences, University of Campania Luigi Vanvitelli, 80131 Naples, Italy; paolo.calabro@unicampania.it; 5Division of Cardiology, Cardiovascular Department, Sant’Anna e San Sebastiano Hospital, 81100 Caserta, Italy; 6Cardiology Division, Azienda Ospedaliero Universitaria di Ferrara, Sant’Anna a Cona, 44124 Ferrara, Italy; cmpglc@unife.it; 7Cardiology Division, Santa Croce e Carle Hospital, 12100 Cuneo, Italy; roberta.rossini2@gmail.com; 8Cardiology-Coronary Care Unit-Cath Lab Division, Azienda Ospedaliera Cannizzaro, 95126 Catania, Italy; simogiub@hotmail.com

**Keywords:** dual antiplatelet therapy, ticagrelor, myocardial infarction

## Abstract

There is uncertainty in cardiologists’ attitudes for prolonging dual antiplatelet therapy (DAPT) with ticagrelor 60 mg beyond 12 months in post-myocardial infarction (MI) patients. We aimed at characterizing the Italian cardiologists’ perceptions and needs in the management of such patients. Two consecutive questionnaires were proposed between June and November 2021, and compiled by 122 and 87 Cardiologists, respectively. Agreement among cardiologists was defined as either a >70% frequency of concordant responses relative to total respondents or following the Delphi method as developed by the RAND Corporation. An agreement was reached on the indication of ticagrelor as the first choice P2Y_12_ inhibitor in MI patients, irrespective of the presentation [ST elevation MI (STEMI), 72%, vs. non-ST elevation MI (NSTEMI), 71%] or the management [invasive vs. conservative (75%)]. A consensus was also achieved on the possibility to consider a patient suitable for long-term DAPT with ticagrelor 60 mg even in case of another P2Y_12_ inhibitor used in the first year after the acute event (74, 85%). To define ischemic and bleeding risks, a consensus was reached on the utilization of one or more scores (87, 71%).

## 1. Introduction

The 2019 edition of the European Society of Cardiology (ESC) Guidelines (GLs) on the so-called “chronic coronary syndromes” (CCS) has partly revolutionized the nosographic approach, yet has also underlined the worrisome morbidity and mortality associated with this condition. Indeed, the clinical presentations of coronary heart disease encompassed by the term were and remain among the most important in terms of cardiovascular burden [1,2,3]. In the latest GLs, the concept of a dynamic disease has emerged, modifying previous terminology referring to “stability” of this condition [4]. In such patients, controlling residual risk is of the utmost importance to avert the danger of recurrent events [5]. To do so, it is necessary to identify the subjects most at risk and to evaluate all relevant therapeutic optimizations, including the duration of dual antiplatelet therapy (DAPT) with aspirin and a P2Y_12_ inhibitor in subsets of patients at high ischemic risk in the absence of a concomitant high hemorrhagic risk [6,7,8]. Both the American College of Cardiology/American Heart Association (ACC/AHA) and the ESC GLs recommend treatment with DAPT based on acetylsalicylic acid (ASA, aspirin) and an oral platelet adenosine diphosphate receptor (P2Y_12_) inhibitor) for 12 months after the acute event, unless a high hemorrhagic risk is present or unless there is a need for surgery that cannot be deferred within this time [9,10]. In recent years there has been also an extensive debate about the optimal duration of DAPT and the best drug combination; however, several studies have shown that in patients with a high-risk profile at baseline, an aggressive treatment approach is associated with better long-term cardiovascular mortality and morbidity [11,12,13].

Translating suggestions from those studies into practice is, however, challenging both for cardiologists (primarily) and healthcare practitioners as a whole. In general, the modalities of mid- and long-term management should consider the risk profile of individual patients, both in terms of initial therapeutic choices and in defining the timing and frequency of controls during follow-up. Indeed, decisions about DAPT prolongation involve aspects that go beyond the perimeter of the usual attention to the atherothrombotic risk profile [14], and involve the cardiologists’ perception on such issues.

To shed light into cardiologists’ opinions on this topic, a group of Italian cardiologists (including interventional and clinical cardiologists, based both in academic and non-academic centers) promoted the initiative of a survey among colleagues to address and discuss the topic, with conclusions drawn applying the Delphi methodology [15,16].

## 2. Subjects and Methods

This study was undertaken between June and November 2021. The topics covered by the survey were developed by the study authors, and presented to a panel of responding clinicians in the form of two consecutive questionnaires [Appendix A] consisting of variously structured questions: some of these involved a response based on a scale ranging from one (maximum disagreement) to nine (maximum agreement); in other questions, the expected responses were dichotomous (Yes/No) or included three possible answers.

In detail, first, an Advisory Board of the five authors, considered experts in this area, was convened and, based on a thorough review of the literature, prepared a first questionnaire (Q1) that included questions on physicians’ demographics, type/size of institutions where they were working, years of experience in the field, and geographic area of their facilities. Some of these data were used as covariates (respondent’s age, years of involvement in the management of cardiovascular disease, and scope of their work). In addition, centers involved were classified with respect to their characteristics. The Advisory Board then met to discuss the results of the Q1 survey and developed a second questionnaire (Q2) to resolve issues that were not clear from Q1 and to better assess the appropriateness of some diagnostic/therapeutic/management procedures.

Q1 and Q2 were administered, through a digital platform, to an Expert Panel (EP) consisting of interventional, outpatient, and inpatient cardiologists recruited from a national database, and ultimately comprising 61 Italian centers [Appendix A] selected randomly and distributed throughout the country. Due to the random procedure in the center selection, the sample is considered representative of a good mix of Italian cardiologists, with different extraction and expertise, comprising both academic and non-academic centers. Q1 was sent out to the EP in August 2021, and statistical analysis performed on 122 responses (75% of the initial database of cardiologists). Q2 was sent to all Q1 respondents in October 2021; responses were collected by November 2021, and statistical analyses here performed on 87 respondents (71% of the original Q1 respondents). Q2 allowed the refinement of some topics that had generated ambiguous responses in Q1.

Responses were analyzed as frequencies relative to total respondents, with 70% selected as the threshold for defining agreement. The agreement among experts was further assessed following the Delphi method as developed by the RAND Corporation [17]. Delphi is a well-established and validated consensus-building process for developing agreement and making group-based decisions in various fields [15,18,19,20]. Traditionally based on the three core concepts of anonymity, controlled feedback, and statistical group response, the method is routinely used in health research and clinical challenges [21]. The method requires the use of a scale ranging from one (maximal disagreement) to nine (maximal agreement), with five corresponding to a neutral opinion about any given item. Scores provided by the experts were then statistically elaborated to obtain an appropriate “index of consensus”. According to The RAND/UCLA Appropriateness Method User’s Manual, Inter-Percentile Range Adjusted for Symmetry (IPRAS) scores, which are a measure of score dispersion adjusted for panel symmetry, were used to determine the level of agreement for each item. The rationale is that, when ratings are symmetric, the Inter-Percentile Range (IPR) required to label an indication as disagreement is smaller than when they are asymmetric. Asymmetry was defined as “the distance between the central point of the IPR and the central point of the 1–9 scale, i.e., 5”. Since the more asymmetric the ratings are, the larger the IPR required to say that there is disagreement, the following mathematical function was developed: IPRAS = IPRr + (AI * CFA), where IPRr is the IPR required for disagreement when perfect symmetry exists; AI is the Asymmetry Index; and CFA is the Correction Factor for Asymmetry. A statement or indication is rated with a disagreement if IPRi > IPRASi. Based on IPR and IPRAS computation it is possible to classify each statement with the appropriateness of a given diagnostic/therapeutic strategy in the following categories: Appropriate (panel median of 7–9, without disagreement); Uncertain (panel median of 4–6 or any median with disagreement); Inappropriate (panel median of 1–3, without disagreement).

## 3. Results

### 3.1. Study Population

One-hundred and twenty-two expert cardiologists represent the study population participating in the project. The mean age of the respondents was 42 years (SD = 9), and mean time of experience in the management of cardiovascular disease was 15 years (SD = 8). The second questionnaire was filled in by 87 cardiologists (72% of the original study population), whose main characteristics were comparable with Q1 respondents (Table 1). Overall, 56 (46%) cardiologists replied to be working in a catheterization laboratory; 50 (41%) in a coronary/intensive care unit or a cardiology ward, and 16 (13%) in an outpatient service (Table 1). The vast majority of the respondents [89 (73%)], replied to work in a public hospital, whereas only 14 (12%) were in a private hospital and 8 (7%) in an out-of-hospital cardiology service.

Table 2 shows the main topics of the study and the indexes of appropriateness evaluated according to the RAND/UCLA Method [complete list in Appendix A].

### 3.2. In-Hospital P2Y_12_ Inhibitor Management

For ST-segment elevation myocardial infarction (STEMI) patients, the consensus was met in the indication of ticagrelor as the first choice among P2Y_12_ inhibitors (88 respondents, 72%), followed by clopidogrel and prasugrel [with 17 (14%)] each (Figure 1A). The choice regarding the P2Y_12_ inhibitor decided in the STEMI setting was primarily made by interventional cardiologists (66, 54% of cases), followed by clinical cardiologist (48, 39%).

A consensus among responders also indicated ticagrelor as the first choice P2Y_12_ inhibitor in NSTEMI patients (86 respondents, 71%), while clopidogrel was selected in 14 (11%) and prasugrel in 22 (18%) (Figure 1B). The same ranking was shown for acute coronary syndrome patients managed conservatively, with ticagrelor utilized in at least 75% of the cases by 48 (55%) respondents, whereas prasugrel was used by only 5 (5%, Figure 1C).

In most cases, there appeared to be no standardized internal protocol for the P2Y_12_ inhibitor choice (88, 72% consensus met). The indication, when appropriate, to prolong DAPT after one year was said to be reported in the discharge letter in 57% of the cases.

No consensus was met on the impact of the possibility to prolong DAPT by the choice of the P2Y_12_ inhibitor at the time of the index event (54% no impact, 46% with an impact). A consensus was met, however, regarding the possibility to consider a patient for long-term DAPT with ticagrelor 60 mg even in case of another P2Y_12_ inhibitor used in the first year after the acute event (74, 85%).

### 3.3. Follow-Up of CCS Patients and Long-Term DAPT Prescription

Even in the midst of the COVID-19 pandemic, half of the respondents (52%) reported to work in structures still without a telemedicine program aimed at the follow-up of CCS patients. The decision to prolong DAPT in the long-term with ticagrelor 60 mg was reported to be carried out by cardiologists working in a hospital (56% interventional cardiologists, 50% cardiologists discharging the patient, 63.1% cardiologists during an outpatient visit). Only in 9% of the cases was the decision carried out by cardiologists working outside the hospital.

### 3.4. Identification of Long-Term DAPT Candidates

In order to define ischemic and bleeding risk, a consensus was reached on the utilization of one or more scores (87, 71%), mainly DAPT (56, 46%) and PRECISE-DAPT (43, 35%) scores (Figure 2). In particular, a consensus was met within cardiologists with less than 10 years of experience to utilize the DAPT score to evaluate ischemic risk in patients who were candidate for long-term DAPT. Older cardiologists (>40 years) tended to utilize the GRACE score significantly more than younger ones (19% vs. 2%). As to the evaluation of bleeding risk, the main score utilized was the PRECISE-DAPT (66% of respondents), followed by the HAS-BLED score (24%). In particular, a consensus was met among out-patient cardiologists in using the PRECISE-DAPT score (83% of respondents).

Age was perceived per se as a main driver in the decision not to prolong DAPT only in 30% of cases. Respondents did not reach a consensus regarding the prevalence of one risk on the other (59% ischemic versus 41% bleeding) in the choice of prolonging DAPT. More than two-thirds of respondents declared that they usually prolonged DAPT after one year in their patients (83, 68%). Mean age of responders was significantly lower for those choosing the shorter DAPT regimen compared with those choosing the longest one (39 vs. 47 years of age).

### 3.5. Outpatient Management at the One-Year Visit

A CCS patient was defined as either: asymptomatic or with stabilized symptoms (median seven, IQR 4, IPRAS 4.60, appropriate); or a patient with elective revascularization at least 12 months before (median 7, IQR 4, IPRAS 4.60) or free from events 12 months after an MI (median 8, IQR 3, IPRAS 6.10). Out-patient service cardiologists reached a consensus regarding the need for a suggestion to prolong DAPT in the discharge letter of candidate patients (median 8, IQR2, IPRAS 6.85, appropriate). CCS was considered a dynamic disease (median 9, IQR 2, IPRAS 7.08).

No single factor emerged as a main barrier in the choice to prolong DAPT after one year. Respondents reached a consensus on the most relevant elements in deciding to prolong DAPT at the one-year visit, namely the favorable balance between ischemic and bleeding risk (median 9, IQR 1, IPRAS 7.60, appropriate) and the absence of adverse events during DAPT (median nine, IQR 1, IPRAS 7.60, appropriate) (Figure 3). Multivessel disease, high-risk clinical features (Diabetes Mellitus, Chronic Kidney Disease, Recurrent Acute Myocardial Infarction (reAMI), Peripheral Artery Disease), and high-risk procedural features (left main disease, three vessel disease, bifurcation stenting, chronic total occlusions, graft treatment, stent length >60 mm), on top of an MI diagnosis, and low bleeding risk, were all considered relevant to identify the ideal candidate for a prolonged DAPT (median 9, IQR 1, IPRAS 7.60; median 9, IQR 1, IPRAS 8.35; median 9, IQR 1, IPRAS 8.35, respectively; appropriate). Responders did not identify one major cause for the underutilization of prolonged DAPT or its withdrawal (Figure 4).

A patient with post-MI CCS, without high hemorrhagic risk, multivesselA patient with post-MI CCS, without high hemorrhagic risk, at high residual ischemic risk by clinical characteristics (diabetes mellitus, chronic renal failure, recurrent acute events, multivessel atherosclerosis)A patient with post-MI CCS, without high hemorrhagic risk, at high residual ischemic risk by procedural characteristics of percutaneous revascularization (main stem disease; three or more lesions; implantation of three or more stents; bifurcation treatment with two stents, treatment of chronic occlusions or venous grafts, total stent length >60 mm)

Responders considered the adherence to international guidelines and consensus documents highly important in the evaluation of DAPT prolongation (median 9, IQR 1, IPRAS 7.60, appropriate), while no consensus was achieved regarding possible actions to implement CCS patients’ follow-up and cure (Figure 5), whereas around 70% of responders attributed importance to the presence of a dedicated outpatient service within the hospital or a more integrate management of the patients between in- and out-of-hospital services. In fact, a large consensus (99%) was achieved in the need of a predefined treatment pathway after MI.

## 4. Discussion

The results of the present Delphi project can be summarized as follows:Ticagrelor is still considered as the first choice P2Y_12_ inhibitor in ACS patients, irrespective of the presentation (STEMI vs. NSTEMI) or the management (invasive vs. conservative);Prolonged DAPT regimens after one year are acknowledged as important strategies by more than two-thirds of respondents, and scores to estimate ischemic and bleeding risks were declared to be frequently utilized in clinical practice;When clinically indicated, the switch from one P2Y_12_ inhibitor to another does not represent a barrier to prolong DAPT with ticagrelor 60 mg.

The choice regarding which P2Y_12_ inhibitor to use in an ACS setting is largely based on the results of large randomized clinical trials that showed the superiority of prasugrel and ticagrelor compared with clopidogrel [22,23]. However, a recent smaller randomized clinical trial showed superiority of prasugrel over ticagrelor in a head-to-head comparison by significantly reducing the rate of MI [24], which led to a Class IIa Level of Evidence B indication in NSTE-ACS guidelines for prasugrel utilization in patients undergoing PCI [25]. In our survey, ticagrelor still appeared to be the first choice for all ACS patients irrespective of the clinical presentation (STEMI or NSTEMI) and management (invasive or conservative). While the indication regarding the conservative management could have been anticipated since prasugrel can be used only when coronary anatomy is known, reasons behind ticagrelor choice in the entire spectrum of ACS clinical presentations are probably multiple: (i) in PLATO, ticagrelor showed a significant reduction in cardiovascular death [23], whereas in TRITON-TIMI 38 prasugrel reduced the incidence of the composite ischemic endpoint with an overall neutral effect on cardiovascular death compared with clopidogrel [22]; (ii) an effect of ticagrelor on cardiovascular death was shown in PEGASUS-TIMI 54 trial [7]; (iii) the combined number of patients enrolled in PLATO [23] and TRITON-TIMI 38 [22] trials (32 232) is eight-fold larger than that in the ISAR-REACT 5 trial (4 018) [24]. In summary, it appeared that expert cardiologists can interpret and weigh results of new evidence and guidelines in the context of all the available data [26].

A thought-provoking finding of the present Delphi project is the reported confidence of the interviewed cardiologists in prolonging DAPT after one year in clinical practice in more than two thirds of the cases. This probably reflects real-life data that show the higher prevalence of patients eligible for ticagrelor 60 mg long-term DAPT in real-life populations when compared to randomized clinical trial ones [27,28].

Despite no consensus being reached regarding the relevance of ischemic and bleeding risks in the clinical decision-making for long-term DAPT prescription, the clinical benefit (ischemic 59%) was valued more than the risk of adverse bleeding events (41%). Indeed, a favorable balance between ischemic and bleeding risk reached a consensus as the most relevant element in the decision to prolong DAPT, and this was confirmed by the fact that 2/3 of the respondents declared to prolong DAPT after one year in their patients. This shows that the interviewed representatives of the cardiology community factored, for their decision, the results of several studies showing that coronary heart disease is dynamic rather than static, and that in patients with a high-risk profile at baseline a more aggressive therapeutic attitude is associated with better long-term cardiovascular outcomes [11,29]. Interestingly, younger cardiologists seemed to be more sensitive to bleeding risk by choosing more frequently shorter (≤1-year) DAPT regimens. In fact, the mean age of responders was significantly lower for those choosing the shorter DAPT regimens compared to those choosing the longer one (39 versus 47 years). This could be due to greater attention paid to identifying patients at high bleeding risk, as suggested by recent position papers [30,31].

A somehow unexpected result is represented by the high rate of scores utilization in estimating the trade-off between ischemic and bleeding risk. This would mirror cardiologists’ good confidence with such tools, at least in the hospital environment. It is important to point-out that even if scores can be useful supporting tools for a comprehensive clinical evaluation, their use has not been demonstrated to improve patients’ outcome. In fact, it has been argued that optimal clinical and procedural care to reduce overall bleeding and ischemic risks should be practiced independent of patients’ scores [32]. Overall, DAPT and PRECISE-DAPT were the most utilized scores [32,33]. A great variability in the utilization of scores according to the risk assessed and the cardiologist’s age were reported. DAPT was the most utilized score for ischemic risk evaluation, while PRECISE-DAPT was the most widely used score for bleeding. This is somewhat surprising, since both scores balance the ischemic vs. the bleeding risk. At the same time, DAPT is a score appositely created to evaluate the benefit of prolonging DAPT, and for this reason may be the one most utilized for ischemic risk; conversely, PRECISE-DAPT is more focused on the trade-off risk evaluation between shorter (3–6 months) vs. longer (12–24 months) DAPT regimens, and for this reason could be the one preferred in patients where the main goal is to assess the bleeding risk [32,33]. Interestingly, at least in part, older cardiologists tended to use older scores, such as the GRACE score, for ischemic risk.

After evaluating the trade-off between hemorrhagic and ischemic risk, which is the most important aspect in deciding to prolong or not to prolong DAPT beyond 12 months, the ideal candidate for a long DAPT was voted to be the patient at high ischemic risk as a result of clinical or procedural characteristics.

The need for a greater collaboration between in- and out of-hospital services with regard to long-term DAPT decision-making is well demonstrated by how outpatients service cardiologists deemed it important to receive an appropriate indication/suggestion to prolong DAPT already in the discharge letter, whereas the implementation of this practice is still suboptimal [34]. This need does not necessarily represent the willingness to rely on another professional’s judgment. This is also reflected by the willingness to switch from another P2Y_12_ inhibitor to ticagrelor 60 mg after 12 months when indicated, even if a shift between ticagrelor 90 mg and 60 mg is certainly easier for the cardiologist performing the 12-month follow-up visit. In addition, interviewed cardiologists showed that their decision about the longer-term DAPT at 12 months is driven more by a careful evaluation of the trade-off between ischemic and bleeding risk, as well as by the absence of adverse events in the first 12 months (tolerability), rather than by a therapeutic inertia based only on following another professional’s decisions. This demonstrates a “dynamic” assessment of a single patient’s risk throughout the follow-up.

## 5. Limitations

We recognize several limitations in this report. First, the overall sample size is small, however, the random selection of 61 participating centers across the entire country and ensuring a good mix of cardiologists with different expertise and working environment should be reassuring with regard to good representativeness of the Italian cardiologists mostly working in a hospital environment. We also indeed recognize, as a second limitation, that the results of the present project should be interpreted in the context of the cardiologists’ sample interviewed, with most of the respondents working in hospital services. The actual process related to long-term DAPT prescription is still hospital-centered, at least in the environment of the interviewed population. Therefore, the point of view, inputs and needs of out-of-hospital cardiologists might be underestimated in the present project. Third, not all responders to the first questionnaire filled in the second questionnaire, further limiting the sample size. Fourth, this analysis is based on self-reported respondents’ assertions, and these may differ from the ones actually enacted (e.g., score utilization and long-term DAPT prescription in two thirds of patients may be overoptimistic estimates).

## 6. Conclusions

The present Delphi project showed that, in a contemporary cohort of expert cardiologists, ticagrelor is still considered the first-choice P2Y_12_ inhibitor in ACS patients; prolonged DAPT regimens and scores aimed at the evaluation of the trade-off between ischemic and bleeding risk seem to be frequently utilized in clinical practice; ischemic and bleeding risks appear to have the same importance in the clinical decision-making; a greater collaboration between in- and out-of-hospital services and predefined pathways for post-MI patients are warranted to guarantee that a higher percentage of patients transitioning to CCS who theoretically can benefit from long-term DAPT actually receive it. 

## Figures and Tables

**Figure 1 jcm-11-02091-f001:**
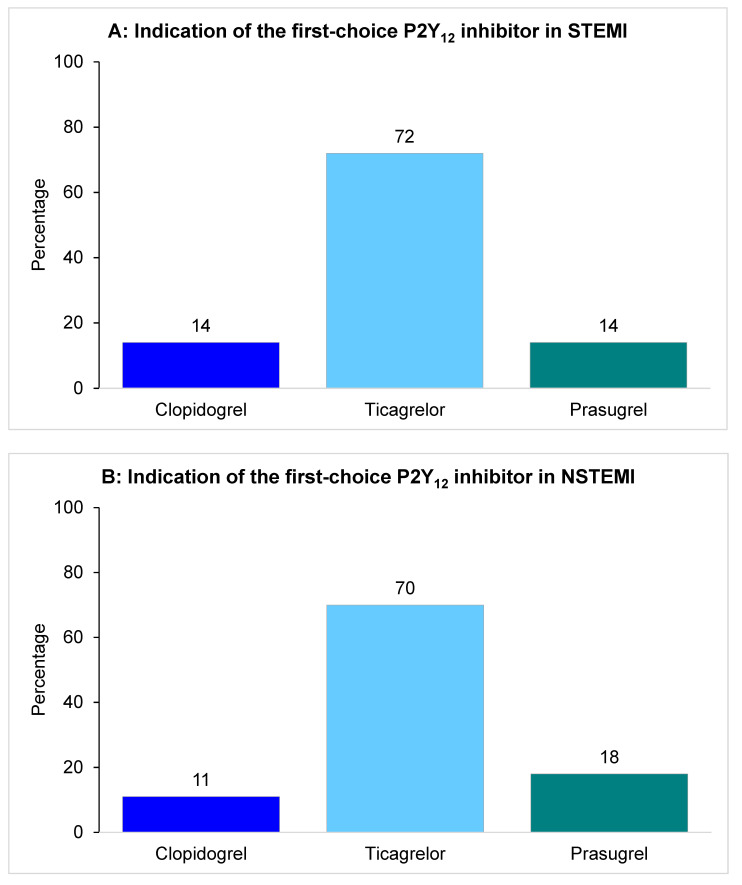
Respondents’ indication of the first-choice P2Y_12_ inhibitor in all acute coronary syndrome clinical settings. (**A**): Indication of the first-choice P2Y_12_ inhibitor in ST-segment elevation myocardial infarction (STEMI). (**B**): Indication of the first-choice P2Y_12_ inhibitor in non ST-segment elevation myocardial infarction (NSTEMI). (**C**): Indication of ticagrelor percentage of use in NSTEMI conservatively managed.

**Figure 2 jcm-11-02091-f002:**
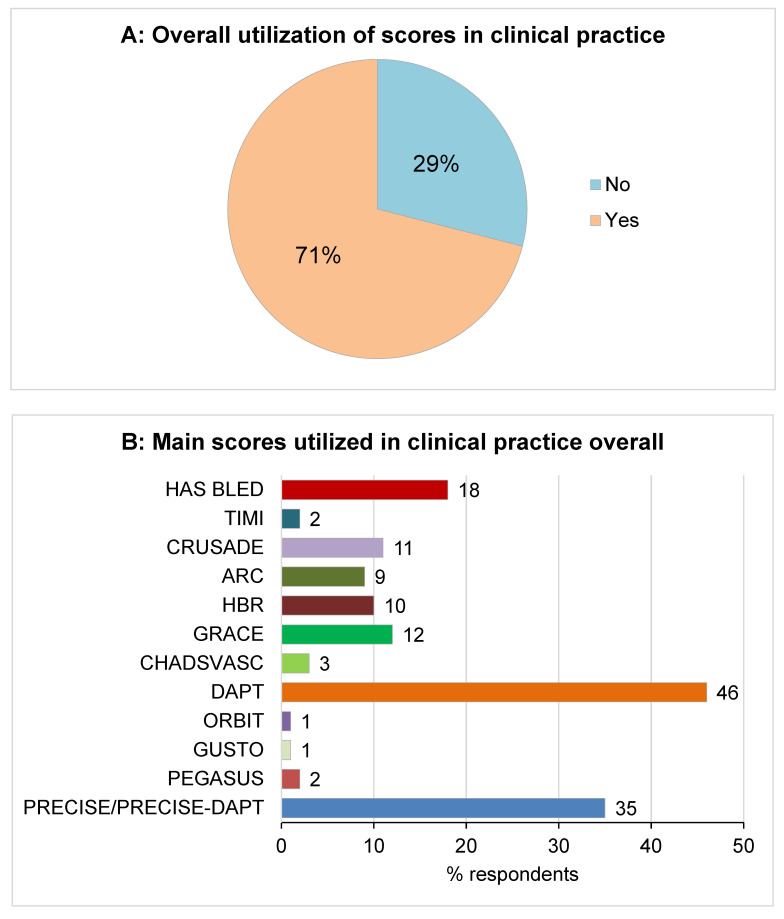
(**A**): Overall utilization of scores in clinical practice. (**B**): Main scores utilized in clinical practice overall. (**C**): Main scores utilized in clinical practice to estimate ischemic risk (%). (**D**): Main scores utilized in clinical practice to estimate bleeding risk (%).

**Figure 3 jcm-11-02091-f003:**
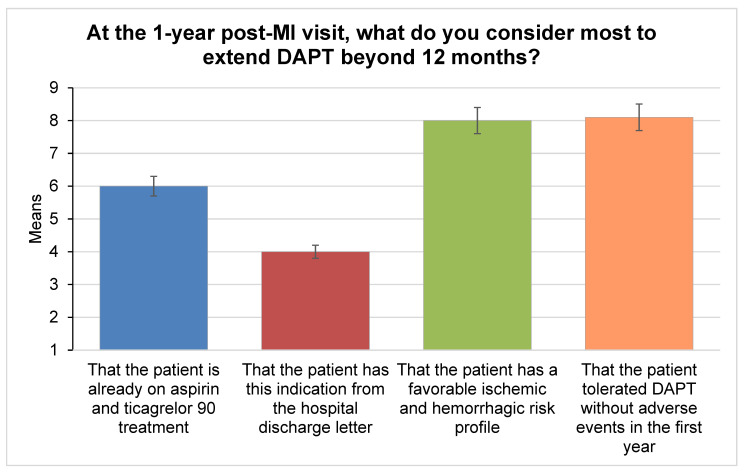
Most relevant elements considered in order to prolong DAPT at 12 months.

**Figure 4 jcm-11-02091-f004:**
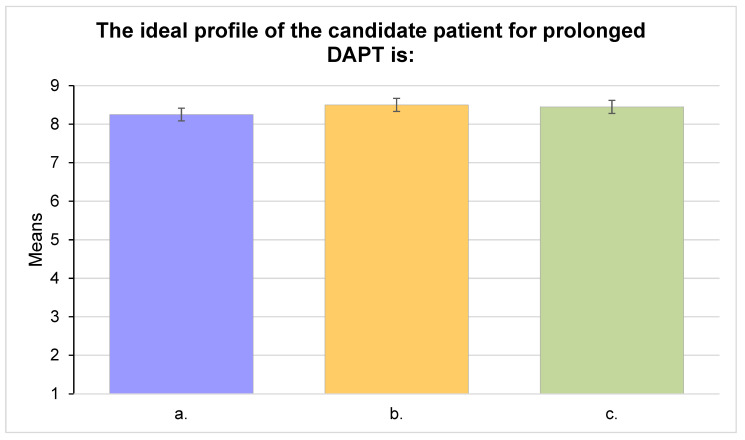
Main barriers to long-DAPT prescription in post-MI CCS patients in clinical practice.

**Figure 5 jcm-11-02091-f005:**
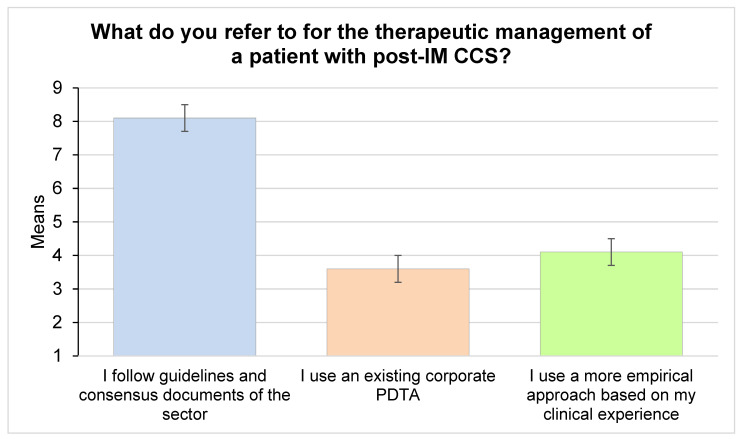
Possible actions aimed at the implementation of follow-up and cure in post-MI CCS patients.

**Table 1 jcm-11-02091-t001:** Characteristics of the study population.

	Q1	Q2
	N = 122	N = 87
Age, years	41.8 ± 8.8	42.1 ± 8.6
Overall working experience, years	16.6 ± 8.4	16.6 ± 8.4
Experience in cardiovascular disease treatment, years	15 ± 8.4	15.1 ± 8.5
*Main occupation*		
Cath lab	56 (46%)	37 (42%)
Coronary Intensive Care Unit/Ward	50 (41%)	38 (48%)
Out-patient service	16 (13%)	12 (11%)
*Place of work*		
University	29 (24%)	
Public hospital	89 (73%)	
Private hospital	15 (12%)	
Hub hospital	40 (33%)	
Spoke hospital	17 (14%)	
Out-of-hospital ambulatory	9 (7%)	

**Table 2 jcm-11-02091-t002:** Appropriateness Indexes evaluated according to the RAND/UCLA Method.

Item	Median	IQR	IPRAS	Assessment
17. How would you define a patient with CCS?				
17.c. Asymptomatic patient (and/or with stabilized symptomatology)	7	4	4.60	appropriate
17.e. Patient with elective revascularization for more than one year	7	4	6.10	appropriate
17.f. Patient more than 1 year after MI with no further events	8	3	6.10	appropriate
18. Do you consider CCS to be a dynamic disease?	9	2	7.08	appropriate
27. At the 1-year post-MI visit. what element do you most consider to extend DAPT beyond 12 months?				
27.c. That he has a favourable ischemic and hemorrhagic risk profile	9	1	7.60	appropriate
27.d. That he tolerated DAPT without adverse events in the first year	9	1	7.60	appropriate
28. The ideal profile of the candidate patient for prolonged DAPT is:				
28.a. A patient with CCS post IM. without high hemorrhagic risk. multivessel	9	1	7.60	appropriate
28.b. A patient with post-MI CCS. without high hemorrhagic risk. at high residual ischemic risk by clinical characteristics (diabetes mellitus. chronic renal failure. recurrent acute events. multivessel atherosclerosis)	9	1	8.35	appropriate
28.c. A patient with post-MI CCS. without high hemorrhagic risk. at high residual ischemic risk by procedural characteristics of percutaneous revascularization (main stem lesions; three or more lesions; implantation of three or more stents; bifurcation treatment with two stents; treatment of chronic occlusions or venous grafts; total stent length >60 mm)	9	1	8.35	appropriate
32. What do you refer to for the therapeutic management of a patient with post-MI CCS?				
32.a. I follow guidelines and consensus documents of the sector	9	1	7.60	appropriate

IQR: interquartile range; IPRAS: Inter-Percentile Range Adjusted for Symmetry.

## Data Availability

Data will be made available to other investigators upon a reasonable request.

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
