# Peer review of "A Therapeutic Pathway in Patients with Chronic Coronary Syndromes: Proposal for Optimization"

_jcm, 2022, doi:10.3390/jcm11082091_

Round 1

Reviewer 1 Report

The authors should be commended for the amount of work that was put into this study.  The study sample size is small and the response between Q1 and Q2 was large.  This would have affected your final results.  I am not familiar with IPRAS when using Delphi methodology, as it is usually accompanied by a Kappa for the final agreement.  However, the authors have detailed the use of the IPRAS.  Means and medians at your percentage response rates and percentage level of agreements would have added scientific significance. I am not convinced that the study adds anything new.

Author Response

See Reply to Reviewers, in the form of a formal letter - attached

Reviewer 2 Report

Interesting study based on the survey . 
The study method is designed properly . 
I have a question on that all the Italian cardiologists are practicing in which level of hospital (eg, tertiary Centre, rural Centre, etc) . Do you also think the guideline based bias can be found in the results ? Advised to describe some more discussions on GDMT . What about the national insurance system on coverage of medication in ACS might influence the use of antiplatelets in this survey ? Overall, this is a good survey and interesting findings . 

Author Response

(The authors gave the same response as above.)
